# "I want to perform and succeed more than those who are HIV-seronegative" Lived experiences of youth who acquired HIV perinetally and attend Zewditu Memorial Hospital ART clinic, Addis Ababa, Ethiopia

**Nahom Solomon**[1]\*, **Mitike Molla**[2], **Bezawit Ketema**[2]

1 Department of Public Health, Mizan Tepi University, Mizan-Aman, Ethiopia, 2 Department of Preventive Medicine, School of Public Health, Addis Ababa University, Addis Ababa, Ethiopia

\* nahomsolomon83@gmail.com

## Abstract

### Background

In Ethiopian Human Immunodeficiency Virus (HIV) prevention program one of the focusing areas is prevention of mother-to-child transmission and decreasing morbidity and mortality among those who already acquired it. However, the needs and the sexual behavior of children who acquired HIV perinatally was not given due attention. Therefore, we conducted this study with the aim of exploring the lived experiences of youth who acquired HIV perinatally to contribute to HIV prevention and control program.

### Methods

We conducted a qualitative study using a phenomenological approach from March to May 2018 among 16 purposively selected youth who were infected with HIV vertically and receive ART services at Zewditu Memorial Hospital, Addis Ababa, Ethiopia. They were recruited based on their willingness after obtaining written informed consent and assent. Data were audio taped, transcribed verbatim in Amharic and later translated to English, and coded using Open Code version 4.02 software. Findings were summarized under four themes by applying interpretative phenomenological analysis.

### Findings

Seven males and nine females, aged 16 to 22 years have participated in the study. These youth reported as they had support from families and ART clinics, while pill-load, and fear of stigma are some of challenges they have faced, especially majorities don't want to disclose their status because of fear of stigma and discrimination. Half of them have ever had sexual relations usually with a seronegative partner and most of these had their first sex in their 17-18 years of age. Unsafe sex was common among them where four girls reported to have had unprotected sex with their seropositive or seronegative sexual partners. Most wish to

**Data Availability Statement:** All relevant data are within the paper and its Supporting Information files.

**Funding:** The authors received no specific funding for this work.

**Competing interests:** No authors have competing interests.

**Abbreviations:** ALHIV, Adolescents Living With HIV; AIDS, Acquired Immune Deficiency Syndrome; ART, Antiretroviral Therapy; ARV, Antiretroviral Drug; EDHS, Ethiopian Demographic Health Survey; HIV, Human Immune Deficiency virus; PAHIV, Perinatally Acquired HIV; PLWH, People Living With HIV; SSA, Sub-Saharan Africa; WHO, World Health Organization.

have purposeful life and love mate of the same serostatus but also fear they may remain alone.

## Conclusion

Youth who had acquired HIV from parents are challenged due to their serostatus and were not sure what type of life they may have in the future. They were also not comfortable in disclosing their serostatus and also engaged in unsafe sexual relation. This calls for an urgent intervention among HIV infected youth and their families; health care providers, and young people in general to halt HIV transmission. Special attention should be given on sexual behavior of all young people (10–24) and in disclosure of HIV status to children and life skills education to cop-up with stigma and discrimination.

## Introduction

The Human Immunodeficiency Virus (HIV), which causes Acquired Immunodeficiency Syndrome (AIDS) is one of the major global public health challenges that has affected the lives of many people especially in low and middle income countries [1]. AIDS is the second leading cause of death among adolescents in Africa. Although mortality due to it has been decreasing globally but age disaggregated data indicated that death among young population, particularly youth is not yet declining and apparently it has tripled [2, 3]. In Ethiopia, 722,248 people lived with HIV, and there were 22,827 new HIV infections and 14,872 people died from HIV/AIDS in 2017 [4].

Perinatally HIV infected youth means a person who acquired the HIV virus from parents and remained HIV positive for the rest of his/her life. Therefore it is a separate epidemic and needs to be handled and managed separately. The advent of puberty and its associated cognitive, emotional and physiological development, coupled with the emergence of sexual expression and the desire to experiment, general risk-taking, high levels of mobility, instability and change, often lead to heightened vulnerability and become sexually active, they carry huge risk of transmitting HIV to others [5, 6] and may even result in a second generation of children to which HIV is transmitted vertically [7].

Age of sexual debut, types of relationships and sexual networking patterns, condom use, contraceptive-use patterns, pregnancy, and recreational drug or alcohol use are significant risk factors in this population. Knowledge, skills, self-efficacy, societal norms and values, gender, sexual aggression and coercion impact their ability to reduce their HIV risk [6]. However mostly they are not properly informed about their status and fail to shoulder the responsibilities expected of them and takes care of themselves and others as well. Despite having different supportive mechanisms and sources of supports, Perinatally HIV infected youth have multiple issues that needs special attention; of these, most reported challenges which include: fear of disclosure, stigma and discrimination, drug load and side effects, economical and psychosocial problems [8–14].

One of the strategies to prevent HIV/AIDS transmission involves targeting persons with chronic HIV infection to reduce their death and disability levels by using antiretroviral therapy and enabling partial immune reconstitution [6]. Related to that, although many works have been done including on Prevention of Mother To Child Transmission (PMTCT); HIV among youth who had acquired the virus through MTCT and who cover about 21% of HIV positive

people in Sub Saharan Africa (SSA) did not get due attention in terms of research. Available studies are mostly limited epidemiological studies which limits understanding of their lived experience and challenges which will help to address the problem at large [7].

Giving attention to youth with perinatally HIV infection has a double benefit for improving their health status and preventing secondary and sexual transmission of HIV, so it is important to generate evidence using proper method and this study was conducted to explore and understand the lived experiences of these population groups. Learning the lived experiences helps to identify the areas where to focus for promoting their health and empowering them to play their role in prevention of HIV transmission. Furthermore it informs policy makers, the community, health care providers and stakeholders to make evidence based intervention.

## Methods

### Setting

The study was conducted from March to May 2018, at Zewditu Memorial Hospital, in Addis Ababa, Ethiopia. Addis Ababa, which has 4 million population, is the capital city of Ethiopia and the diplomatic center of Africa [15]. The city has 128,912 people living with HIV, 4,221 newly HIV infected people and 1,955 died from HIV/AIDS in 2017 [4]. This study was conducted at Zewditu Memorial Hospital which is the first ART service delivery hospital since 2003 and has grown to become the largest HIV care and treatment hospital in the country [16]. In 2017, the hospital served 7125 HIV positive people including 496 youth (238 males and 258 females) [16]. The hospital was selected for this study because of its long time ART service experience and high client flow; especially of youth.

### Study design and sample

Qualitative study with hermeneutic phenomenological approach was applied [17]. Youth aged 15–24 who acquired HIV vertically and were on ART follow up at least for a year, those who could give written consent and assent, and those who can use Amharic language were considered to be eligible for the study. Sixteen youth who volunteered to take part in the study were recruited with the assistance of nurse from the ART clinic. The principal investigator, who is also the data collector had made discussion with the nurse on how to select youth for the study and it was done by two phases. Firstly the nurse contact youth while they come for collecting their drugs, asks them if they are willing to participate in the study and refers to the data collector only those who gave consent. Secondly, the data collector gives detail information of the study and asks for written consent and assent, then did interview with those who gave consent and assent till the information get saturated.

### Data collection

In depth interviews were conducted using open ended interview guide. Prior to the actual data collection, pretest was done to learn about the process of interview, content, time it takes and necessary amendments were made based on the interview. All data were collected by the principal investigator.

The interviews were conducted in the hospital at private room that could ensure privacy. During the interview; participants were given space to talk about themselves and their experiences as a person living with HIV. In the process, questions and probes focusing on their concerns of life, support mechanisms and challenges they had faced, their sexual behavior and experiences were explored. The interview took an average of 42 minutes and repeated contact was done when more explanation was needed. Throughout the interview, in addition to audio

recording, the PI who conducted the interview has also taken note about their facial expressions and other nonverbal reflections.

### Ensuring trustworthiness

In establishing credibility; the investigator took adequate time with study participants and since the interviewer was almost at close age with the youth, it helped to develop rapport. All interviews were audio recorded and kept for cross checking if needed. Audit trial, debriefing and feedback from colleagues and co-authors were used in managing the data. The inquire process and findings were described in detail so that any reader of the report will be able to use and researchers may replicate the study at other similar settings.

### Data analysis

Code book was developed prior to data collection and amendment was done during and after data collection was completed. Analysis of data started and was done simultaneously with data collection. Information stored on audio recorder was listened repeatedly and transcribed verbatim by the interviewer and translated to English language. Preliminary analysis was done to see for saturation of the information and emerged theme. After compiling all translated word documents; coding and categorization were done using Open Code version 4.02 software [18].

Following interpretative phenomenological analysis principles [19, 20]; the lived experiences of participants and meanings they gave to their words were considered in coding and categorization. Emerged themes were interpreted with the four fundamental lifeworld existential of; lived body, lived space, lived time and lived relationship [21]. Under the umbrella of IPA, thematic analysis was followed to summarize the findings. The quotations which reflect the majorities' view and unique once were used.

### Ethical approval and consent to participate

The study was approved by the Research and Ethics Committee of the School of Public Health, Addis Ababa University and Addis Ababa Health Bureau, Public Health Research and Emergency Management Core Process. Written informed consent was obtained from youth 18 and above years old and care givers of under 18 youth. Assent was obtained from youth who were under 18 years of age. All study activities were abided by basic ethical principles of; respect for autonomy, beneficence, non-maleficence, confidentiality and justice to maximum level.

### Findings

Following socio demographic characteristics, the major findings of the study are summarized under four themes, which includes; (1) Reported health status and HIV/AIDS lessons, which involves; reported health status, feelings about youth HIV status, life plan and role of youth in the prevention of HIV, (2) Sources of supports of youth: which involves; family and clinical care and social support, (3) Challenges of youth: which covers; threat to future life, experience of taking ARV drugs, disclosure of their HIV status, stigma and discrimination, (4) Sexual relation experiences of youth: which encompasses; youth's sexual relation experience, and youth's sexual relation interest. All of these are elaborated under different sections below.

### Socio demographic characteristics of study participants

Sixteen youth who acquired HIV vertically had participated in this study. All were Christian religion followers, aged between 16 and 22 years and nine were females. While one of them has learnt up to grade 8 and the other one is advanced diploma holder, 14 were students attending

their studies at secondary (n = 6) and tertiary levels (n = 8). Eight of them reported to live with their biological parents, while the rest live with either close relatives or foster families.

## Theme-1:-Reported health status and lessons learned from youth about how they feel about their HIV/AIDS status

This is one of the themes which cover the following three categories.

### Category-1:-Youth's perceived health status

Most reflected that, they used to feel anxious during their first periods of knowing their status, but later they took it easy and most have started to take ART in their early age; they also reported as they are healthy and confident enough to perform as anyone does. One of the participants described his feelings about his health as follows:

*"I feel healthy and I think I can perform anything that can be performed as that of any healthy person" (19 years old male participant)*

Some have reported experiences of minor illnesses; especially before initiation of the ARV drug, and three reported sever health problems (anemia) during their ART follow up period.

### Category-2:-Youth's feeling about their HIV status

Majorities have reported as they don't like to think about their HIV status; some revealed that they feel disturbances and loneliness when they think of their HIV status. Those who expressed such feelings have also reported to have stress and frustration. One of the participants indicated this situation as follows:

*"You know it is punishment without our fault, so things happening not due to our fault disturb us." (18 years girl)*

*Again another participant reported the following feeling*

*"There is a feeling I feel as human being; I mean I need to live being healthy as any one, at the beginning I felt as the virus is posted on my face (yehone viresu fitih lay yeteletefe hulu new yemimeslh), I mean I feel as all know my status; What makes me to shock is people's response" (20 years old male participant)*

Most of the participants have reported different emotional reactions when they understood that they had HIV. Some reported to have cried, shocked, isolated themselves and delayed to accept it, but later on they convinced themselves and start all over again. Some hope to get cure from God. Their lived human relation was limited to very close families; most reported that only very few people knew their status even among families. Even one boy reported that he didn't ever disclose his status for his father and takes his drug even hiding him. He did that not to tension his father because his father himself is HIV positive, lost many things including his wife and currently lives alone, and he thought as his son is HIV negative.

### Category-3:-Youth's life plan

Almost all study participants have depicted that they wish to be successful in their life. They want to show that they are capable of doing everything. Some of them dream to be a famous actress and film director, psychiatrist, psychologist, engineer, pilot and continue their

education and obtain a PhD and be influential person in their area of specialization. An under-graduate student reflected her plan as follows:

> *"I want to reach to a better level. Even I want to be a better person than those who are not infected with HIV. If I achieve that, I will get the chance to be an examplenary to others." (19 years girl)*

*Another study participant has also discussed her wish as follows:*

> *"I want to be psychologist; I want to work on prisoners and students. I want to work on forgotten people. You know when you close and talk them; you get what you didn't expect. They want to share their ideas but they don't get who hear them. So I will be happy if I reach them and give them what can I do." (21 years old girl)*

### Theme-2:-Sources of supports of youth who acquired HIV perinatally

This theme encompasses the following major categories.

### Category-1:-Family care

Family is the major source of support revealed by most of the study participants. As indicated above, almost all study participants live with their biological families and extended families; even the one who lives with fosters also see them as parents and feel as she is with her father and mother. Accordingly most participants reported that they received care and support from their families including extended families but some of them have also reported as they did not get adequate support. The following responses show both sides:

> *"I am living with my aunt; She has also HIV, we are living supporting each other and all my families know my status, and support me" (19 years girl)*

> *"I am living with my uncle (father's brother) since 5 years back and he doesn't care of me. His principle is 'live your own' and doesn't ask me even whether I take the drug or not and of my living." (19 years girl)*

### Category-2:-Clinic care

All of the study participants were following their ART service at a unit where they started and it was with respective age based service. Almost all indicated that they had good support and benefitted a lot from the clinical services. Almost all of them have long time follow up experience in the hospital and they are familiar with the environment very well. They have close relation with clinical staffs including doctors, nurses and social workers.

> *"My doctors treat me in a sisterly and brotherly manner; I also receive a paternalistic approach from the older clinicians" (18 years boy)*

Youth were also comfortable with the opening hours of the health facility which in their terms was flexible. The fact that they were able to pick their pills at any time during working hours including Saturday was highly appreciated by them as it allows them to pick their pills after classes.

Acknowledging the good services of the clinic; there were also issues which were suggested to be seen and corrected. Some have reported presences of problems with time management, the service being merely of prescribing drug than addressing their psychosocial issues and no

enough laboratory services. Again it was reflected though the clinic appointment is flexible; still some youth reported missing of class due to clinic appointment.

### Category-3:-Social support

Youth club is the major supportive group reported by most participants. It is an edutainment program where youth who had acquired HIV from parents get together every two weeks and learn, play, entertain, share different things and get friends. The youth club is divided in to two; one for those who are under 15 years of age and the other for those who are above 15 years of age. Although all don't participate due to their personal and family reason, those who participate in it reflected as it is good for them and love it.

> *"The youth club has many benefits. I get friends, I feel happy when I meet with someone whom I considered is mine, I talk with them freely than healthy guys, I mean, I feel more comfortable and know that I am not alone. in the club, no one points a finger on anyone, I always wait eagerly for this day to come and meet with my friends" (19 years old girl)*

Beyond the youth club; some had supports from their school and local friends, and some teachers. They have reported that their close friends understand them and treat them very friendly.

### Theme-3:- Challenges of youth who acquired HIV perinatally

Challenges of youth were discussed in detail and the following categories were the most focusing areas

### Category-1:-Threat of future Life

Majority of them revealed they have concerns of their future life relative to; succeeding in education, getting job and forming marriage. They have discussed in detail raising different points as indicated below.

> *"What concerns me is my life; I mean my future life; I am a human being and when I grow and reach at some level I will form a family. . . but sometimes I feel like I may remain alone; You know, the community don't believe that we are able to do anything as any other healthy person." (19 years old girl)*

> *"You know I am female; actually not only females, everyone should form a marriage, So such thing frighten me. Even by now when my peers have boyfriend I can't do that. Males ask me but I don't want because I know what I have. I don't want to enter to unwanted things by disclosing myself. It is a threat for my future. How to get a husband? I need child but it doesn't be, it worries me." (17 years old girl)*

### Category-2:-Fear of stigma and discrimination

Almost all study participants have discussed that they are concerned about how people treat them and the place given to them in the community. They reflected that though there is a general perceptio as there is a change in stigma and discrimination, youth indicated that there is still a huge problem with regard to awareness of HIV and attitude to people living with HIV. Youth reported that they have no freedom to do what they wish freely.

*"In day-to-day activities you may meet people and the way people view HIV is not good, It is good if community's awareness is changed. There is a general belief that attitude of people towards people living with HIV is changed but still it is not enough, it is not yet there,. . . . . . there are shocking things; people say 'a person with HIV is stunted, unable to walk, bed ridden and the like'. . ..." (18 years old girl)*

Other study participants have also reflected on issue of community's awareness as follow:

*"I will be happier if we speak freely (benetsanet bininager des yilegnal), you know expressing our internal feeling, if people's attitude change and if we play equally with them as anyone, I will be happier" (17 years old female participant)*

*"Our community is not changed well. For example If you stay around waiting room for a minute you see people (PLWH) wearing eyeglass, covering their face, and trying to hide themselves, which is lack of confidence; it is said the community is changed but it is not changed." (22 years old female participant)*

Youth have also articulated that community's negative attitude and awareness about HIV/ AIDS and people living with HIV made them feel isolated and limit their freedom, which altered their lived human relation. The following statement depicts this:

*"Awareness of most people is somewhat poor; actually it is better than formerly but still much has to be done in the area, they view HIV as something different bad and while children most of us didn't bring HIV by ourselves; we inherited from parents; they don't think that, if you see in our family; my father's wife (stepmother) doesn't have a good feeling about me" (22 years old girl)*

## Category-3:-Disclosure of HIV status of Youth who acquired HIV perinatally

Disclosure has two dimensions: the first is disclosure of HIV status of the youth by parents or care givers to the youth themselves and the second is disclosure of their HIV status for others.

### Disclosure of HIV status of the youth for themselves

Most appreciated the fact that their HIV serostatus was disclosed by their caretakers for which they have reported as beneficial though some reported that they were crying as the situation was shocking and they also said believing what they have heard was difficult; But also reported as their life has changed positively after they knew their HIV status. They indicated that the time after disclosure was special and a turning point in their life where they started to give due attention to themselves including drug adherence and care. Most of them reported to have known their status around their 14 years of age. Some of the study participants knew their status incidentally when they visit clinic for other medical services or by urging their parents to tell them the reasons for taking the pills every day.

Participants were also asked to suggest the best time for disclosure of HIV status for children about their HIV status and most suggested it should be around 15 years of age. The reason for choosing this age was reflected as follows:

*"If they knew before they are aged 15 years; they may not understand the issue very well and even they may disclose their HIV status to others unnecessarily. But if it is in their 15^{th}, they could be capable of handling everything and they will take care of themselves" (20 years boy)*

### Disclosure of their HIV status for others

Disclosing HIV serostatus to others is not easy for most participants. Youth believed that, disclosure of their serostatus may predispose them for rejection, stigma and discrimination by people around them. So they don't support to disclose their status to others. They believed the fact that they did not disclose their status for others has benefitted them very much, because they said that they do everything they need freely including engaging in sexual relations without any threat of people's judgment, stigma and discrimination.

*"Many times not merely at work place even at school; if it is known as one has HIV it is difficult to live, work and learn together, so I will be very happy if such bad attitude and thought at school and anywhere be corrected." (19 years girl)*

*"To tell you the truth I have a boyfriend. . . . . ., he didn't know my status, I didn't tell him, and I didn't try to openly talk." (18 years old girl)*

### Category-4:-Experience of taking ARV drugs

Most youth reported that taking ART every day which has bad taste, side effects and pill load is exhausting. Some also reported that forgetting to take the pills is common

*"I wish if the drugs load is at minimum, again most drugs are big in size and difficult for handling. It is good if sweet taste is added on them as well." (22years old girl)*

*"When I swallow the pill, it gives me headache. Sometimes, I don't need to pass my time having headaches." (20-years boy)*

One of the common problems they reported was related to challenges to privacy when taking the ART pills. Almost all reported that they take their drugs secretly in private area; they do not want to take it when people are around them. To avoid this some of them reported that there are times when they do not stick to their schedule of ART in terms of time or they may miss a day. The following quote indicates how youth are challenged to maintain their ART taking as scheduled:

*"I take it (the ART pill) hiding from others, for example in a bus I turn myself to the window, cover myself with my backpackand and take my drug. . . . . . my mother usually tells me that as I shouldn't give chance for people to see me when taking the pill." (19 years old boy)*

Because of challenges they face, youth in this study wish to have a treatment which will be taken infrequently. While some reflected the rumour they have about new drug development as follows:

*"The current drug is difficult, however, I heard about a new drug which is in the form of an injection that can be taken once every six month. I will be happy if I get that" (20 years boy)*

### Theme-4:-Sexual relation beliefs and experiences of youth who acquired HIV perinatally

This was another major focusing area when discussing the lived experience of youth and the following categories were derived from the discussion.

## Category-1:-Beliefs about sexual relations

There was a difference in belief about a good age to have a sexual debut for a young person. While some preferred to delay until they get a job and when they are aged in their late twenties some indicated that, they are okay any time if they meet with the right partner. Most want to have relationships and marriage with a seropositive person like them. However, they also indicated that they will also accept if they get a proposal from a seronegative person who can accept them as a seropositive person. Almost all study participants believe in disclosing their HIV status before starting sexual relation.

> *"For the future it is God who knows. We are human, we try. I don't sit saying I don't love. I should not hurt that person and he shouldn't also. I will be happy if we are similar (both HIV positive) but if not; just if he has awareness and accepts me that is also not bad to be with HIV negative" (19 years old girl)*

## Category-2:-Sexual relation experiences of youth

The sexual life of the youth is not different from what they believe in as indicated above. Half of the study participants reported that they ever had a boyfriend or girlfriend and didn't have any sexual engagement. The rest have ever had boyfriend or girlfriend. Two of those (one female and one male) who had had mates faced rejection due to their serostatus; their friends were HIV negative and were not willing to continue with them. A female study participant who faced rejection described the situation as follows:

> *"Once I was in a relationship with a seronegative boy. He did not know my status, we stayed together for a long time and when our relationship became strong I was afraid, at the end I told him but his response was very bad, he rejected me" (19 years old girl)*

Four female study participants reported that they had unsafe sexual intercourse with their boyfriends. All reported that they have disclosed themselves before having a sexual contact. Three out of the four girls reported to have started having sex when they were 17 years of age and the other one was 18 years old. While two of the girls have their first sex with HIV positive partner, one had it with a sero-negative person. The fourth one was engaged with a person whom she did not know his serostatus without condoms. Regarding condom usage; though they have knowledge to use and believe in it; they ignored it due to their partners' unwillingness. One of the study participants who engaged in sexual contact with her HIV positive boyfriend reported that:

> *"We were at fire age and we did it for satisfying our sensation. . ... we stayed for three months in one home like husband and wife. . .we did it (sexual intercourse) without condom. . . . . .initially we didn't make any care; he(partner) said 'since we are the same (HIV positive) there is no need to do any care(no need to use condom)'." (21 years old girl)*

Another girl indicated that she accepted the suggestions of her partners when it comes to condom use and sex because she loved them (both the current, who is HIV positive and the ex-boyfriend, who didn't know his serostatus).

> *"I didn't hide him anything, I told him all and he believed in me. Therefore, it is difficult to say no for the one who believed in you. . . .. We didn't use condom. . . what is advised is to use*

*condom,. . .it has no problem for me to use condom but I should consider the feelings of others too. Hence, since he didn't like it(condom) we didn't use." (19 years old girl)*

## Discussion

In this study we found that, youth who acquired HIV through vertical transmission knew their status at an average of 14 years ranging between 10 and 17 years of age. The youth appreciated that knowing their status at age 15 is preferable as they are sort of matured to accept the burden and also helps them to protect others from HIV. Youth in this study also indicated that thinking about their status usually results in bad feelings and they prefer to ignore it and hope the future will bring innovative medicine and they also leave their fate to God. Youth do not want to disclose their status to others because of fear of stigma and discrimination. Boredom from pill load, bad taste, searching for privacy to take the pills, getting a partner who has similar sero-status are some of the concerns and challenges they had. It is revealed that almost half have engaged in sexual relation which is unsafe; even with a sero-negative partner.

All participants of this study have reported a benefit from knowing their HIV status and strongly believed that children should learn their status timely. Knowing self-status helped them to feel more responsible and take care of their and others health too. This finding is similar with studies from Botswana, South Africa, and Puerto Rico where similar population groups have depicted the same self-disclosure timing and benefit from it [7, 22–24]. When it is said self-disclosure it is important to make it in plan than leaving children to learn about their HIV status incidentally when they visit health care facilities for other medical services or leave them to learn by their own or hearing from somebody else.

Study participants indicated that they had different reactions when their status was disclosed for them by caretakers. Some reported crying, shocking and not believing what they have heard. this finding is consistent with studies from United Kingdom, Kenya, USA and South Africa [7, 24–27], where similar population group reflected the same reactions to their self-disclosure. However those who used to suspect and collect information about HIV/AIDS earlier in their long time ARV follow up didn't conveyed serious emotional reactions.

Regarding the sources of supports of youth with perinatally acquired HIV; families and care givers take the lion share. Studies from Botswana and South Africa also reported same finding that family care is an important source of support for adolescents with perinatally acquired HIV [22, 28]. The health care services and the service providers' approach were other major sources of supports reported by almost all study participants. Especially the opening hours of the facility including Saturdays were very important for youth. This finding is consistent with studies report from Kenya, Sweden and Ethiopia [26, 29, 30]. This implies that if the health care services are accessible enough and accompanied with improved friendly services, they will produce tangible health outcome among such population group which helped them to adhere with the ART regimen.

Youth club, which is an edutainment club of youth with perinatally acquired HIV where they get together every two weeks and learn, share ideas, play together and is an opportunity for mixing with youth of similar status and a way of satisfaction and sharing their experience. At the same time, it is a new avenue for them to get their soulmates. Others also found out that these clubs are very important to give youth hope and happiness. For example, care givers of children with perinatally acquired HIV, from Addis Ababa and Oromia Regional State, have also depicted that their children have liked and benefited from peer supporting group [30]. Another study from Botswana also revealed similar program in which adolescents with perinatally acquired HIV enjoy and appreciate such programs [22].

Youth in this study have faced different challenges where some of the challenges related to medication (the pill load, unpleasant taste, side effects, life-long treatment and privacy) has affected their adherence. Care givers of children with perinatally acquired HIV in other studies from Ethiopia also reported their children are tired of taking the drug [30]. Similarly adolescents with HIV from South Africa have also reported their interaction with friends at home and school is affected by medication, they complained that being on medication makes their lives very difficult [28].

Disclosure to others is one of the major concerning and challenging issues of youth living with perinatally acquired HIV. As pointed out above almost all appreciated to know their status timely but most of them reported that except for very few close friends, they don't need to disclose their status for others. Similarly a study from South Africa reported as such population group are reluctant to disclose their status to others [24]. Another study in Ethiopia among care givers of children with perinatally acquired HIV reported experience of stigma and discrimination from people and for that reason majorities have reported to hold their status in secret [30].

Stigma and discrimination is the major concerning issue identified in this study and similar findings were reported from Kenya and South Africa [8, 10, 12]. The 2016 Ethiopian Demographic Health Survey (EDHS) has also found that there are a discriminatory attitudes where 48% of women and 35% of men thought that children living with HIV should not be able to attend school with children who are HIV negative, while 55% of women and 47% of men would not buy vegetables from a shopkeeper who has HIV [31].

On the other hand, in this study, youth claimed that they got the disease not by choice but from their parents and they felt that the world is unfair to them. This is alarming for a new HIV epidemic as youth in this study reported to have unsafe sex even with seronegative partner, this finding is supported by reports from systematic review from Sub Saharan Africa [32].

Half of the participants did not engage in sexual relations and those who had engaged are already experiencing risky sexual relation including early sexual debut. The fact that most do not use condoms could predispose them to both unwanted pregnancy and sexually transmitted infections. They believe that there is no risk in having unsafe sex with a person who has same sero-status is dangerous and needs a lot of focus, similar finding was reported from South Africa and Uganda [11, 33]. The reasons for the girls who have unsafe sex with both sero-positive and negative partners was mainly love and keeping the feelings of their partners. However, there is a sense of subordination of girls where they cannot decide on their safety and fertility. This population as indicated above is a special population who is desperate and uninformed about the risks of unsafe sex; hence, if left unchecked this will be one of the sources of new HIV epidemic.

Generally, this study explored the lived experiences of youth who had acquired from parents and identified the focusing areas. The findings will be an input for further studies and strategies to take actions for improving the health care services of such population groups and further more empowering them and the community, the care providers and agents to control and prevent HIV transmission.

This study would have been more informative if the experiences of parents and care givers, health care providers and other stakeholders were included. The other limitation could be the fact that the study was conducted in health care facility might have influenced participants' responses and there might be social desirability bias. However, the fact the interviewer was the PI supervised by a senior expert helped to get reliable responses though extensive probing and data scrutiny during analysis.

## Conclusion

Youth in this study face challenges related to stigma and discrimination, rejection, boredom from the ART regimen and non-adherence. On the other hand, youth do not want to disclose their status because of stigma and discrimination and engaged in early sexual debut and have unprotected sex with both seronegative and seropositive people. This is an alarming issue in the prevention of HIV transmission. The fact that these youth are alienated and believed that they did not get HIV by choice is dangerous for further HIV transmission if not checked timely.

## Supporting information

**S1 Fig. Theme tree.**
(TIF)

**S1 Table. Code book.**
(DOCX)

**S1 Data. 5 of Transcribed data.**
(ZIP)

**S1 File. Study participant Information sheet.**
(DOCX)

**S2 File. Interview guide.**
(DOCX)

## Acknowledgments

We are grateful to the study participants for their kind cooperation and Zewditu Memorial Hospital ART clinic staffs. We acknowledge Addis Ababa University, the College of Health Sciences and colleagues from Behavioral Health Sciences Unit School of Public Health for their contribution in one way or another.

   **Declarations:** We declare that this thesis is our original work, has never been published and that all the resources and materials used for this study are recognized and cited, and people who involved in are acknowledged.

## Author Contributions

**Conceptualization:** Nahom Solomon, Mitike Molla, Bezawit Ketema.

**Data curation:** Nahom Solomon.

**Formal analysis:** Nahom Solomon.

**Investigation:** Nahom Solomon.

**Methodology:** Nahom Solomon, Mitike Molla, Bezawit Ketema.

**Supervision:** Nahom Solomon, Mitike Molla, Bezawit Ketema.

**Writing – original draft:** Nahom Solomon.

**Writing – review & editing:** Nahom Solomon, Mitike Molla, Bezawit Ketema.

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
