## [Decision Letter · Decision Letter 0]

11 Mar 2021

PONE-D-20-23565

“I want to perform and succeed more than those who are HIV-seronegative ” Lived Experiences of youth who acquired HIV Perinetally and attend Zewditu Memorial Hospital ART clinic, Addis Ababa, Ethiopia.

PLOS ONE

Dear Dr. Solomon,

Thank you for submitting your manuscript to PLOS ONE. After careful consideration, we feel that it has merit but does not fully meet PLOS ONE’s publication criteria as it currently stands. Therefore, we invite you to submit a revised version of the manuscript that addresses the points raised during the review process.

The manuscript has been evaluated by two reviewers, and their comments are available below. You will see the reviewers have commented on the interest and relevance of your study. However, the reviewers have also raised critical concerns and the manuscript will need significant revision before it can be considered for publication – you should anticipate that the reviewers will be re-invited to assess the revised manuscript, so please ensure that your revision is thorough. I have outlined some of the key concerns noted by the reviewers below, but you should respond to all concerns mentioned by the reviewers in your response-to-reviewers document. 

The key concerns noted by the reviewers relate to the clarity of the reporting of the methods and results. Specifically, the reviewers requested additional information about participant recruitment and the interview procedures, including the semi-structured interview guide. Additionally, Reviewer 1 noted that the relationship between participant sociodemographic characteristics and the study findings should be discussed. 

We look forward to receiving your revised manuscript.

Kind regards,

Danielle Poole

Academic Editor

PLOS ONE

Journal Requirements:

4. In the Methods, please describe how the questionnaire was validated and/ or pretested. If this did not occur, please provide the rationale for not validating the questionnaire.

Reviewers' comments:

Reviewer's Responses to Questions

**Comments to the Author**

1. Is the manuscript technically sound, and do the data support the conclusions?

Reviewer #1: Yes

Reviewer #2: Yes

2. Has the statistical analysis been performed appropriately and rigorously? 

Reviewer #1: N/A

Reviewer #2: N/A

3. Have the authors made all data underlying the findings in their manuscript fully available?

Reviewer #1: Yes

Reviewer #2: No

4. Is the manuscript presented in an intelligible fashion and written in standard English?

Reviewer #1: Yes

Reviewer #2: No

5. Review Comments to the Author

Reviewer #1: It is a study about a relevant theme, it is well developed and well elaborated.

It presents some absences in the methodology and inconsistencies in the results that need to be clarified, described below.

The object under study is clear and justified and the method chosen is appropriate to the objectives of the study.

The methodological path is lacking information that gives it more transparency. It is necessary to clarify better how the recruitment of participants was done. It is reported that the study was carried out with patients at a hospital specializing in HIV treatment. It is reported that 16 young people volunteered to take part in the study. How did this happen? How did they find out about the study and why did they volunteer? Was there prior knowledge between the interviewer and the participants? Were there refusals to participate?

It is said that the interviews followed a semi-structured script, however the content of that script was not presented. It is said initially that the interviews were collected by the principal investigator and then that the interviewer was young like the interviewees, which is inconsistent with the initial information. After all, is the principal investigator as young as the participants? Were all interviews conducted by the same investigator?

It is said that facial expressions and other non-verbal manifestations were noted, however, nothing is said in the results about this.

In the results, the sociodemographic data were not associated with any category, therefore, they do not contribute anything to the study. Also, nothing was informed about participants’ social and economic status. There was no questioning regarding the participant's gender, education, family composition or religion, data that were cited. For example, were there any differences between the experiences of boys and girls? And among those who had family support or not? Were there differences according to education?

In the discussion there is an excerpt on page 20, in the second line of the second paragraph, that young people acquired HIV from their parents or the community in general. So, can vertical transmission come from the community at large? On the same page 20, in the sixth and seventh lines of the third paragraph, an unsupported statement by the results is made: “The reasons for the girls who have unsafe sex with both seropositive and negative partners was mainly love and keeping the feelings of their partners ”.

Reviewer #2: Thank you for the opportunity to review this great work! I appreciated learning more about young people's experiences in Ethiopia.

Although I believe this work is important, there are several items which should be addressed, which prevented me from recommending it for publication.

I've provided my specific comments below:

Please review the spelling, grammar, and word choice in the manuscript. I acknowledge that this is a difficult ask, but some grammar issues interfered with the ability to understand the writing (e.g. the way perinatal was described, the use of don't rather than do not, the term "youth" inconsistently).

I would have appreciated understanding more about the semi-structured guide and the phenomenological approach was applied. This should be described in more detail in the methods. It would be useful to have some theory or frame work for this qualitative analysis.

Similarly, the organization of results should be summarized more concisely. I would recommend moving the quotes into a table and using the results to provide themes/major categories. The results of the study are not clear to me, because each section included only one quote and it wasn't clear to me if these quotes/themes showed up often in the interviews.

6. PLOS authors have the option to publish the peer review history of their article (what does this mean?). If published, this will include your full peer review and any attached files.

Reviewer #1: No

Reviewer #2: No

---

## [Author Response · Author response to Decision Letter 0]

19 Apr 2021

Responses to Reviewers’ questions and comments

Firstly, I would like to thank reviewers and the editor for your comments and constructive suggestions. As you recommended; necessary corrections are made and clarifications are given for each points raised by reviewers and the editor as presented below.

Part-I: Editor’s comment

Comment-1: Please ensure that your manuscript meets PLOS ONE's style requirements, including those for file naming.

Response-1: our manuscript is prepared with the recommended format

Comment-2: thoroughly copyedit your manuscript for language usage, spelling, and grammar.

Response-2: we have gone through the manuscript and made corrections as needed.

Comment-3: Please include additional information regarding the survey or questionnaire used in the study and ensure that you have provided sufficient details that others could replicate the analyses. For instance, if you developed a questionnaire as part of this study and it is not under a copyright more restrictive than CC-BY, please include a copy, in both the original language and English, as Supporting Information.

Response-3: We didn’t develop or adopt a questionnaire, we only used interview guide and detail interview was made by probing based on participants’ response.

Comment-4: In the Methods, please describe how the questionnaire was validated and/ or pretested. If this did not occur, please provide the rationale for not validating the questionnaire.

Response-4: Again since we didn’t use a questionnaire, there was no need of validation process. But prior to the actual data collection, pretest of the interview guide was done to learn about the process of interview, content, time it takes and necessary amendments were made based on the interview.

Comment-5: Your ethics statement should only appear in the Methods section of your manuscript. If your ethics statement is written in any section besides the Methods, please move it to the Methods section and delete it from any other section. Please ensure that your ethics statement is included in your manuscript, as the ethics statement entered into the online submission form will not be published alongside your manuscript.

 Response-5: We moved the ethical statement to method section.

Part-II: Reviewr-1

Comment-1: It is necessary to clarify better how the recruitment of participants was done. It is reported that the study was carried out with patients at a hospital specializing in HIV treatment. It is reported that 16 young people volunteered to take part in the study. How did this happen? How did they find out about the study and why did they volunteer? Was there prior knowledge between the interviewer and the participants? Were there refusals to participate?

Response-1: The principal investigator, who is also the data collector had made discussion with the nurse on how to select youth for the study and it was done by two phases. Firstly the nurse contacts youth while they come for collecting their drugs, asks them if they are willing to participate in the study and refers to the data collector only those who gave consent. Secondly, the data collector gives detail information of the study and asks for written consent and assent, then did interview with those who gave consent and assent till the information get saturated. Based on that, sixteen youth who volunteered to take part in the study were recruited with the assistance of nurse from the ART clinic.

Comment-2: It is said that the interviews followed a semi-structured script, however the content of that script was not presented.

Response-2: Sorry it is a write up error and what we used is interview guide/unstructured question.

Comment-3: It is said initially that the interviews were collected by the principal investigator and then that the interviewer was young like the interviewees, which is inconsistent with the initial information. After all, is the principal investigator as young as the participants? Were all interviews conducted by the same investigator?

Response-3: Yes, all data were collected by the principal investigator and although he was not as young as participants, relatively he was close to them in his age and related things (28 by the time of data collection).

Comment-4: It is said that facial expressions and other non-verbal manifestations were noted, however, nothing is said in the results about this.

Response-5: of course note was taken and indicated in the transcription but not found to be that much significant to discuss in the manuscript.

Comment-6: In the results, the sociodemographic data were not associated with any category, therefore, they do not contribute anything to the study. Also, nothing was informed about participants’ social and economic status. There was no questioning regarding the participant's gender, education, family composition or religion, data that were cited. For example, were there any differences between the experiences of boys and girls? And among those who had family support or not? Were there differences according to education?

Response-6: the background of participants is presented in the first section of the finding report and of course we didn’t focus on discussing associations, because the aim was just to describe experiences but we have also presented experiences of youth relative to their background like, who have family support or not, who have challenges related to job, relationship, school and friends. 

Comment-7: In the discussion there is an excerpt on page 20, in the second line of the second paragraph, that young people acquired HIV from their parents or the community in general. So, can vertical transmission come from the community at large?

Response-7: what was intended to say was that ‘community in general doesn’t have good attitude to HIV positive youth who acquired it from parents’, so sorry for confusing you and we have corrected it.

Comment-8: On page 20, in the sixth and seventh lines of the third paragraph, an unsupported statement by the results is made: “The reasons for the girls who have unsafe sex with both seropositive and negative partners was mainly love and keeping the feelings of their partners ”.

Response-8: this is supported by the finding and even there is a quote which clearly describes this. So please consider that.

Part-III: Reviewer-2

Comment-1: Please review the spelling, grammar, and word choice in the manuscript. I acknowledge that this is a difficult ask, but some grammar issues interfered with the ability to understand the writing (e.g. the way perinatal was described, the use of don't rather than do not, the term "youth" inconsistently).

Response-1: Thank you for your comments and we have tried to make necessary correction throughout the manuscript.

Comment-2: I would have appreciated understanding more about the semi-structured guide and the phenomenological approach was applied. This should be described in more detail in the methods. It would be useful to have some theory or frame work for this qualitative analysis.

Response-2: Sorry for the write up error and what we used is unstructured question or open ended question. As indicated in the document, the study followed a phenomenological approach and interpretative phenomenological analysis.

Comment-3: Similarly, the organization of results should be summarized more concisely. I would recommend moving the quotes into a table and using the results to provide themes/major categories. The results of the study are not clear to me, because each section included only one quote and it wasn't clear to me if these quotes/themes showed up often in the interviews.

Response-3: we feared putting all findings in a single table may make it congested and less attractive to read; otherwise, we added quotes which are more commonly reflected and even unique responses as well.

---

## [Editor Report · Decision Letter 1]

5 May 2021

“I want to perform and succeed more than those who are HIV-seronegative ” Lived Experiences of youth who acquired HIV Perinetally and attend Zewditu Memorial Hospital ART clinic, Addis Ababa, Ethiopia.

PONE-D-20-23565R1

Dear Dr. Solomon

We’re pleased to inform you that your manuscript has been judged scientifically suitable for publication and will be formally accepted for publication once it meets all outstanding technical requirements.

Kind regards,

STELLA REGINA STELLA TAQUETTE, Ph.D.

Guest Editor

PLOS ONE
---

## [Editor Report · Acceptance letter]

17 May 2021

PONE-D-20-23565R1 

*“I want to perform and succeed more than those who are HIV-seronegative ” Lived Experiences of youth who acquired HIV Perinetally and attend Zewditu Memorial Hospital ART clinic, Addis Ababa, Ethiopia.*

Dear Dr. Solomon:

I'm pleased to inform you that your manuscript has been deemed suitable for publication in PLOS ONE. Congratulations! Your manuscript is now with our production department. 

Kind regards, 

on behalf of

Professor STELLA REGINA STELLA TAQUETTE 

Guest Editor

PLOS ONE